# Disentangling the Information in Species Interaction Networks

**DOI:** 10.3390/e23060703

**Published:** 2021-06-02

**Authors:** Michiel Stock, Laura Hoebeke, Bernard De Baets

**Affiliations:** KERMIT, Department of Data Analysis and Mathematical Modelling, Ghent University, 9000 Gent, Belgium; laura.hoebeke@ugent.be (L.H.); bernard.debaets@ugent.be (B.D.B.)

**Keywords:** information theory, species interaction networks, diversity, effective numbers

## Abstract

Shannon’s entropy measure is a popular means for quantifying ecological diversity. We explore how one can use information-theoretic measures (that are often called indices in ecology) on joint ensembles to study the diversity of species interaction networks. We leverage the little-known balance equation to decompose the network information into three components describing the species abundance, specificity, and redundancy. This balance reveals that there exists a fundamental trade-off between these components. The decomposition can be straightforwardly extended to analyse networks through time as well as space, leading to the corresponding notions for alpha, beta, and gamma diversity. Our work aims to provide an accessible introduction for ecologists. To this end, we illustrate the interpretation of the components on numerous real networks. The corresponding code is made available to the community in the specialised Julia package EcologicalNetworks.jl.

## 1. Introduction

The use of networks to address ecological questions has become increasingly popular [1,2,3]. Ecological networks not only allow for studying species, but, more importantly, also their interactions [2]. Networks can be used to represent all kinds of ecological interactions, such as predation, parasitism, and mutualism, while the composing interactions can indicate energy transfer, exchange of material, or even exchange of information.

Indices borrowed from the field of information theory can be used to characterise the structure of these networks [4]. In ecology, one uses the term ‘index’ for a measure that quantifies diversity. In this work, we will often refer to information-theoretic measures as indices when they are used for this purpose. Many networks show a particular organisation, where interactions are heterogeneously distributed among species [5]. Generalists interact with many species, while specialists interact with only a few species [6]. This combination of generalists and specialists makes a network more robust [7].

Graphs are commonly used to represent the ecological interactions between different species in an ecosystem [8,9]. A graph consists of nodes that are connected through edges. The nodes represent the species, while the edges connect the interacting species. A distinction can be made between unipartite and bipartite graphs. Any two species can interact in a unipartite graph. Therefore, any two nodes can be connected [2]. A food web without distinct sets of species is an example of a unipartite network. A bipartite network, on the other hand, consists of two disjoint groups of species, with the species from the first group only interacting with species from the second group [10], for instance, feeding interactions between two trophic levels. Some specific examples of bipartite graphs are pollination networks, host–parasite networks, seed dispersal networks, and anemone–fish networks. Our work predominantly focuses on bipartite networks. In Figure 1 (left), interactions between five anemone species and five (anemone-) fish species are visualised as a bipartite graph. The two distinct interaction levels are the anemone species and the fish species, respectively, the bottom and top interaction level. This mutualistic symbiosis between anemones and fishes of the genus *Amphiprion*, as observed by Ricciardi et al. [11] in the Manado region of Indonesia, will be used throughout this work as an illustration.

Information theory is part of probability theory and statistics [12]. It can be applied in a variety of settings, including ecology. Entropy, which is the key measure of information theory, was initially proposed by Shannon in communication theory to study the compression of messages and communication over a noisy channel [13]. Entropy can be used to quantify the expected information content, choice, and uncertainty [14].

Similar to, for example, the English language, ecological interaction networks also have a certain structure, as interactions are not random [15]. Measures from the field of information theory can be adapted to analyse these interaction patterns [16]. A few years after Shannon formulated the basic principles of information theory, MacArthur [17] applied it to ecological networks. He used information theory to analyse the stability of ecosystems by computing the entropy of the energy transfers in food webs. The more energy pathways present in the food web, the higher the uncertainty of the energy flow and the higher the network’s stability. The amount of choice, as quantified by applying information theory to the network, can hence be used as a index of stability of a network [17].

Entropy conveys how much information is contained in an outcome and, thus, how surprising a particular outcome is [18]. The more diverse a system, the more uncertain an outcome will be [19]. The more species an ecological community contains or the more evenly the species are distributed, the higher the uncertainty [20]. It will be difficult to predict an interaction in a very diverse ecological community with a lot of equally distributed species. The uncertainty will be lower in a community with only a few species or a few prevalent ones. The Shannon entropy measure, which is also known as the Shannon–Wiener index [21], has become the most commonly used diversity index [19,22]. However, the use of diversity indices, including entropy, has been criticised on many occasions, since applying different indices to the same ecological community has resulted in contradictory outcomes [23]. This has led to several incorrect conclusions, causing some ecologists to mistrust information theory [16]. However, the defective performance of information theory in ecology is not due to the shortcomings of the indices, but it is rather caused by misinterpretation. Entropy can be used to quantify the diversity of a network [4,24], but it is solely an index of diversity and by no means a direct equivalent of diversity [20,25]. Therefore, the proper interpretation of information-theoretic indices is a key factor when analysing ecological networks.

This article provides an overview of the different information-theoretic indices and their ecological interpretation. We have already explored this kind of analysis in Stock et al. [26]. This work is a detailed treatise of this approach. Understanding and interpreting the relationships between the different indices is eased by visualising their values. Graphical representations also aid in ecological interpretation and they enable us to efficiently compare different interaction networks. We present two visualisation methods: bar plots and entropy triangle plots. Barplots are especially useful in visualising the relative importance of the different information-theoretic components of a given interaction network, while the entropy triangle is especially suited to comparing multiple networks. We introduce a conversion to effective numbers to clarify the relation between entropy and diversity, and to prevent misinterpretation of entropy as a diversity index. We illustrate the proposed methodology on several types of ecological networks.

## 2. Information Theory for Species Interaction Networks

### 2.1. Ecological Couplings

To straightforwardly apply information theory, rather than represent the network as a graph, one uses an n×m incidence matrix *Y*. The rows and column of such a matrix represent the species of the two trophic levels. For bipartite networks, the *n* rows represent the species of one interaction level, while the *m* columns represent the species of the other interaction level. An incidence matrix can contain information regarding the frequency or strength of the interactions (i.e., a weighted matrix) or solely indicate the presence or absence of an interaction (i.e., a binary matrix). Binary observations of interactions are more frequently recorded than weighted descriptions of interaction networks [27]. A binary representation could be seen as a loss of information, as every interaction becomes equally important [28]. However, taking the strength of interactions into account can also lead to mistakes, since the observed frequencies do not always reflect the true frequencies. Quantitative observations of interactions strongly depend on the sampling effort [8], and they often result in undersampling [29]. In this work, we opted to illustrate our methods on binary incidence matrices (possibly obtained through binarizing, i.e., mapping non-zero values to one).

Figure 1 (middle) shows the binary incidence matrix *Y* of the bipartite interaction network between anemone species and fish species. The *n* rows and *m* columns of the matrix *Y* represent, respectively, the anemone species (i.e., the bottom interaction level) and the fish species (i.e., the top interaction level). A matrix element $Y_{ij}$ is equal to 1 if the species *i* of the bottom interaction level interacts with species *j* of the top interaction level and it is equal to 0 otherwise. In the incidence matrix shown in Figure 1 (middle), 1 indicates that anemone species *i* is visited by fish species *j*, while 0 indicates the opposite. However, an interaction between two species is not a pure yes–no event, as the interaction may be rare or depend on several local and behavioural circumstances. As such, we follow Poisot et al. [30] and compute the n×m probability matrix *P* of the joint distribution as
(1)$$P_{ij}=\frac{Y_{ij}}{\sum_{k=1}^{n}\sum_{l=1}^{m}Y_{kl}}\;.$$
This value can be interpreted as the probability that species *i* interacts with species *j*.

In our earlier work, we called this normalized incidence matrix an *ecological coupling* [31]. This coupling arises from random and targeted interactions between the species and it is dependent on the relative species abundances.

In the context of mutualistic symbiosis between anemones and fishes, as shown in Figure 1 (right), $P_{ij}$ is the probability that anemone species *i* is visited by fish species *j*. When interactions are associated with the energy transfers between trophic levels, as in food webs, *P* can be interpreted as the probability distribution of the system’s energy flow. The incidence matrix reveals the distribution of the energy flow from the bottom of the network, the energy source, to the top of the network, the energy sink [32].

The marginal distributions of both interaction levels can be computed as
(2)$$p_{i}^{B}=\sum_{j=1}^{m}P_{ij}\;\mathrm{and}\;p_{j}^{T}=\sum_{i=1}^{n}P_{ij}\;,$$
where $p_{i}^{B}$ is the probability that bottom species *i* establishes an interaction and $p_{j}^{T}$ is the probability that top species *j* establishes an interaction. Note that we introduced two random variables, *B* and *T*, for the bottom species and the top species, respectively. The probability matrix *P* can be augmented to indicate the marginal probabilities, as shown in Figure 1 (right). In this matrix, $p_{i}^{B}$ is the probability that anemon species *i* is visited and $p_{j}^{T}$ is the probability that a visit is made by fish species *j*.

### 2.2. Information Theory for Interaction Networks

Given the above probabilistic interpretation, measures that were borrowed from the field of information theory can be applied to characterise interaction networks. Foremost, we recall the concept of *entropy*, which is defined for a random variable *X*, as
(3)$$H\left(X\right)=\sum_{x}p_{X}\left(x\right)\;\log_{2}\frac{1}{p_{X}\left(x\right)}=-\sum_{x}p_{X}\left(x\right)\;\log_{2}p_{X}\left(x\right)\;,$$
where $p_{X}$ is the probability mass function of *X* [13]. By convention, 0·log0 is evaluated as 0 [33].

Therefore, values with zero probability will have no influence [34]. The logarithm to the base two is commonly used [35]. Therefore, we will drop the explicit notation of the base.

When base two is used, all of the information-theoretic measures are expressed in bits [36].

Entropy conveys how much information is contained in an outcome and, thus, how surprising a particular outcome is [18]. When the probabilities of all possible outcomes are equal (i.e., *X* is uniformly distributed), the entropy is maximal, since the effective outcome is the most difficult to guess [34,37]. Imagine the situation where a fish has to choose between a green and an orange anemone. The random variable *X* represents the outcome of the experiment and $p_{X}\left(x\right)$ is the probability that *X* takes value *x*. If both anemone species are equally desirable, then the probability distribution $p_{X}$ is uniform. The probability that the fish chooses the orange anemone is equal to the probability of choosing the green one, namely $\frac{1}{2}$. The entropy is now maximal, since every outcome is equally likely and, thus, equally surprising. When every outcome is of equal probability, we obtain the largest amount of information by observing the outcome of the experiment, since the effect was the hardest to predict. Suppose that the green anemone species would be less desirable, with the probability of being chosen equal to $\frac{1}{8}$. In that case, the entropy is reduced to 0.54 bits, which is less than the maximal entropy of one bit when both anemone species are equally desirable. The probability distribution is no longer uniform when one colour is preferred over the other, since it is much more likely that the fish chooses the orange anemone. The more the distribution deviates from the uniform distribution, the less information we obtain by observing the outcome, since we know better what outcome to expect. The entropy is equal to zero in the extreme case where the probability of selecting the orange anemone would be one. We obtain no new information from observing which colour the fish chooses, since we already knew that the outcome would be orange. We can extend this simple example of one fish choosing an interaction partner to an incidence matrix that represents multiple ecological interactions.

The entropies of the marginal distributions of the bottom species *B* and the top species *T*, the *marginal entropies*, can be computed as
(4)$$H\left(B\right)=-\sum_{i=1}^{n}p_{i}^{B}\log p_{i}^{B}\;\mathrm{and}\;H\left(T\right)=-\sum_{j=1}^{m}p_{j}^{T}\log p_{j}^{T}\;.$$
The *joint entropy* of the bivariate distribution is computed as
(5)$$H\left(B,T\right)=-\sum_{i=1}^{n}\sum_{j=1}^{m}P_{ij}\log P_{ij}\;.$$
The marginal entropies quantify the equality of the species at the bottom and top interaction level, or, in the context of mutualistic symbiosis between anemones and fishes, the equality of the anemone species and fish species, respectively. A large value indicates that the marginal distribution of the species of the interaction level is close to a uniform distribution. In contrast, a low value indicates that some species dominate the interactions more than others. On the other hand, joint entropy can be used to analyse the distribution of the interactions.

When the logarithm to base two is used, the entropy is expressed in bits. In this case, we can interpret entropy as the minimal number of yes–no questions that are, on average, required to learn the outcome of an experiment [38]. For species interactions, this boils down to the average number of questions needed to identify an interaction or interaction partner. The answer to these questions is ‘yes’ (1) or ‘no’ (0), so one bit is needed to store the information. For example, suppose that an ecosystem contains four species (*a*, *b*, *c*, and *d*) that occur with relative frequencies $p_{a}=0.5$, $p_{b}=0.25$, $p_{c}=0.125$, and $p_{d}=0.125$. Because species *a* is most abundant, the first question one might ask to identify a species is “Is it species *a*?”. In the fifty percent of the cases that the answer is ‘yes’, one has identified the species using a single question. If the answer is ‘no’, then one has to ask additional questions. The next natural question would be “Is it species *b*?”. Again, if the answer is ‘yes’, one has identified the species; otherwise, one has to pose a third question. This question could be “Is it species *c*?”, which settles the matter as we were left with only two options (*c* and *d*). Because we can identify *a* using a single question (50% of the cases), *b* using two questions (25% of the cases), and *c* and *d* using three questions (12.5% of the cases each), we can identify the species using an average of 1.75 questions. Given that the entropy of this system equals
(6)−0.5log(0.5)−0.25log(0.25)−0.125log(0.125)−0.125log(0.125)=1.75,
we know that this scheme is optimal. However, if the species would be present in equal proportions, this scheme would no longer be optimal, as it now requires 2.25 questions on average. In this case, a different set of questions, starting with, for example, “Is it species *a* or *b*?”, followed by a question to distinguish between the remaining two options, would be optimal. This scheme always requires two questions. Because the entropy of a uniform discrete distribution on a set of four elements is equal to 2, we know that we cannot improve this scheme. This interpretation of entropy expressed in bits as the average number of questions required to identify the interaction or interaction partner is also applicable to other information-theoretic measures, as presented later in this work.

The *conditional entropy* of *B* given *T*, and vice versa, are defined as
(7)$$H\left(T|B\right)=\sum_{i=1}^{n}p_{i}^{B}\;H\left(T|B=b_{i}\right)=\sum_{i=1}^{n}p_{i}^{B}\;\sum_{j=1}^{m}\frac{P_{ij}}{p_{i}^{B}}\;\log\frac{p_{i}^{B}}{P_{ij}}=\sum_{i=1}^{n}\sum_{j=1}^{m}P_{ij}\log\frac{p_{i}^{B}}{P_{ij}}$$
and
(8)$$H\left(B|T\right)=\sum_{j=1}^{m}p_{j}^{T}\;H\left(B|T=t_{j}\right)=\sum_{j=1}^{m}p_{j}^{T}\;\sum_{i=1}^{n}\frac{P_{ij}}{p_{J}^{T}}\;\log\frac{p_{j}^{T}}{P_{ij}}=\sum_{i=1}^{n}\sum_{j=1}^{m}P_{ij}\log\frac{p_{j}^{T}}{P_{ij}}\;.$$
These measures quantify the average uncertainty that remains regarding the top species when the bottom species is known and the average uncertainty that remains with regard to the bottom species when the top species is known, respectively. In the example of mutualistic symbiosis between anemones and fishes, these measures quantify the remaining uncertainty regarding the fish, respectively, anemone species, given that the anemone species, respectively, fish species, is known. Suppose that, for instance, each fish species visits a single anemone species and that each anemone species is visited by a single fish species. In that case, the marginal entropy of both anemone species and fish species is maximal, since both marginal distributions are uniform. However, the conditional entropy is zero because an anemone species is only visited by a single fish species. There is no freedom of choice. If we know the anemone species, then there is no more uncertainty regarding the interacting fish species, since each anemone species is only visited by one specific fish species. Conditional entropy can also be interpreted as the average number of questions needed to identify an interaction partner, as explained above. When the conditional entropy is zero, there is no freedom of choice and no uncertainty about the interaction partner. Therefore, no questions will need to be asked. A conditional entropy that is different from zero indicates that there is remaining uncertainty [39], thus, freedom of choice, for the anemone species or fish species. In that case, questions are needed in order to identify the interaction partner since there are multiple possibilities.

The specificity of the interactions can be more directly quantified by the *mutual information*:(9)$$\begin{array}{cc} I(B;T) & =H(B)-H(B|T) \end{array}$$(10)$$\begin{array}{cc} \; & =H(T)-H(T|B) \end{array}$$(11)$$\begin{array}{cc} \; & =H(B)+H(T)-H(B,T)\;, \end{array}$$
which is symmetric with respect to *B* and *T*, i.e., I(B;T)=I(T;B), and that always satisfies I(B;T)≥0. Mutual information quantifies the average reduction in uncertainty regarding *B*, given that *T* is known, or vice versa. It expresses how much information about *B* is conveyed by *T*, or how much information regarding *T* is conveyed by *B*. When *B* and *T* are independent, *B* holds no information about *T*, or vice versa; therefore, I(B;T) is equal to zero [40]. Mutual information can be interpreted as a measure of the efficiency of an interaction network [39], as high mutual information implies that the species are highly specialised towards a single or a few ecological partners [41].

Finally, the *variation of information* is defined as
(12)V(B;T)=H(B,T)−I(B;T)=H(B|T)+H(T|B).
This measure is the difference between the joint entropy and mutual information. It is the sum of the average remaining uncertainty regarding the bottom species and top species when, respectively, the top and bottom species are known. In the example of mutualistic symbiosis between anemones and fishes, it is the sum of the average remaining uncertainty about the anemone species when the fish species is known and the average remaining uncertainty about the fish species when the anemone species is known. It captures the residual freedom of choice of the species, and it can be interpreted as a measure of stability [15]. The more redundant interactions, the higher the resistance of the network against the extinction of interaction partners [3]. The variation of information and, thus, the stability of the network, can increase when the number of possible interaction partners of the species increases or when the interactions become more equally distributed, thus increasing the uncertainty.

Rearranging the formula above results in the relation between the joint entropy, mutual information, and variation of information:(13)H(B,T)=I(B;T)+V(B;T).
This formula suggests a trade-off between mutual information (i.e., efficiency) and the variation of information (i.e., stability) for an interaction network with a given joint entropy. The ecological interpretation hereof will be discussed more extensively later in this section.

The information-theoretic decomposition of an interaction network can be visualised in a bar plot [37], as opposed to more misleading Venn diagrams. This bar plot displays the relationships between the joint entropy, the marginal entropies, the conditional entropies, and the mutual information of an interaction network. The variation of information is indirectly represented, as it is the sum of the two conditional entropies. The barplot shown in Figure 2 displays the information-theoretic decomposition of the interactions between the five anemone species and five fish species presented in Figure 1. The contributions of the different components to the joint entropy and their relative importance can be analysed and interpreted using this plot.

The above-defined measures can be linked to a uniform distribution, whose entropy is always maximal. When the interactions are uniformly distributed over the *n* species of the bottom interaction level, every species has the same probability of interaction $p_{i}^{B}$, namely $\frac{1}{n}$. Therefore, when the probability distribution is uniform, the marginal entropy of the bottom species can be computed as
(14)$$H\left(U_{B}\right)=-\sum_{i=1}^{n}p_{i}^{B}\;\log p_{i}^{B}=-n\;\frac{1}{n}\;\log\frac{1}{n}=\log n\;.$$
Similarly, the marginal entropy of uniformly distributed top species can be computed as
(15)$$H\left(U_{T}\right)=-\sum_{i=j}^{m}p_{j}^{T}\;\log p_{j}^{T}=-m\;\frac{1}{m}\;\log\frac{1}{m}=\log m\;.$$
Finally, every interaction is equally likely when the joint distribution is uniform. A network with *n* bottom species and *m* top species comprises nm potential interactions. Therefore, in the case of a uniform joint distribution, every interaction has the same probability $P_{ij}$, namely $\frac{1}{nm}$. Thus, the joint entropy of the uniform distribution can be computed as
(16)$$H\left(U_{BT}\right)=-\sum_{i=1}^{n}\sum_{j=1}^{m}P_{ij}\;\log P_{ij}=-nm\;\frac{1}{nm}\;\log\frac{1}{nm}=\log\left(nm\right)$$
and it is equal to the sum of the two marginal entropies $H(U_{B})$ and $H(U_{T})$.

The differences in entropy between the uniform distributions and the corresponding true distributions are defined as
(17)$$\begin{array}{cc} D(B) & =H\left(U_{B}\right)-H\left(B\right)\;, \end{array}$$
(18)$$\begin{array}{cc} D(T) & =H\left(U_{T}\right)-H\left(T\right)\;, \end{array}$$
(19)$$\begin{array}{cc} D(B,T) & =H\left(U_{BT}\right)-\left(H\left(B\right)+H\left(T\right)\right)=D\left(B\right)+D\left(T\right)\;. \end{array}$$
These measures quantify how much each distribution deviates from the corresponding uniform distribution [42]. Note that the difference for the joint distribution is not equal to the difference beweten the entropy of a uniform bivariate distribution and the joint entropy, but rather to the sum of the marginal differences in entropy. We can see H(B)+H(T) as the joint entropy of the random vector (B,T), while assuming that *B* and *T* are independent. This renders the differences in entropy being additive, while joint entropy is not.

The difference in entropy as compared to a uniform distribution, the mutual information, and the variation of information are related by the following balance equation [42]:(20)$$H\left(U_{BT}\right)=\log\left(nm\right)=D\left(B,T\right)+2\;\;I\left(B;T\right)+V\left(B;T\right)\;.$$
This can be demonstrated by combining the equations shown above:(21)$$\begin{array}{cc} \; & D(B,T)+2\;\;I(B;T)+V(B;T) \end{array}$$(22)$$\begin{array}{cc} & =D(B)+D(T)+2\;\;I(B;T)+V(B;T) \end{array}$$(23)$$\begin{array}{cc} & =H\left(U_{B}\right)-H\left(B\right)+H\left(U_{T}\right)-H\left(T\right)+2\;\;I\left(B;T\right)+H\left(B;T\right)-I\left(B;T\right) \end{array}$$(24)$$\begin{array}{cc} & =H\left(U_{B}\right)+H\left(U_{T}\right)=\log n+\log m=H\left(U_{BT}\right)\;. \end{array}$$
The balance equation can be decomposed into the separate contributions of the marginal distributions of the bottom and top species:(25)$$\begin{array}{cc} H(U_{B}) & =\log n=D(B)+I(B;T)+H(B|T)\;, \end{array}$$(26)$$\begin{array}{cc} H(U_{T}) & =\log m=D(T)+I(B;T)+H(T|B)\;. \end{array}$$
Note that this equation also illustrates why the term I(B;T) occurs twice in the global balance equation. These equations show how the maximal potential information of an ecological network is divided into a component expressing that some species are more important or active than others (D), a component that is related to the specific interactions between species (I) and a final component comprising the remaining freedom of the interactions (V). Table 1 presents an overview of these components of the decomposition, for the marginal distributions as well as the joint distribution.

A ternary entropy diagram or entropy triangle plot can be used to visualise the different components of the balance equation. Each side of the triangle corresponds to one of these three components. The entropy triangle enables a direct comparison of different networks, since each network will be represented by a single dot in the triangle. Such a diagram can be constructed for the total balance equation, as well as for the marginal balance equations of the bottom and top interaction level. Figure 3 displays these three entropy triangles. In order to determine the location of a network in the triangle, the balance equation is normalised by dividing all the components of the equation by the entropy of the corresponding uniform distribution [42]. For the triangle of the joint distribution, the computation of the three coordinates of a network is based on the following normalised equation:(27)$$\frac{H(U_{BT})}{H(U_{BT})}=\frac{D(B,T)}{H(U_{BT})}+\frac{2\;\;I(B;T)}{H(U_{BT})}+\frac{V(B;T)}{H(U_{BT})}=1\;.$$
Recall that $H(U_{BT})=\log nm$. Each term of this sum corresponds to the coordinate of the network on one of the three sides of the triangle. Normalising the components of the balance equation by the maximal entropy results in values that are between zero and one that can be plotted on the triangle. The same applies for the balance equations of the marginal distributions:(28)$$\begin{array}{cc} \frac{H(U_{B})}{H(U_{B})} & =\frac{D(B)}{H(U_{B})}+\frac{I(B;T)}{H(U_{B})}+\frac{H(B|T)}{H(U_{B})}=1\;, \end{array}$$(29)$$\begin{array}{cc} \frac{H(U_{T})}{H(U_{T})} & =\frac{D(T)}{H(U_{T})}+\frac{I(B;T)}{H(U_{T})}+\frac{H(T|B)}{H(U_{T})}=1\;. \end{array}$$
A prime is added to the corresponding symbol to denote the normalised component, as used in the entropy triangle. For the total balance equation, this results in:(30)$$D^{\prime }\left(B,T\right)+2\;\;I^{\prime }\left(B;T\right)+V^{\prime }\left(B;T\right)=1\;.$$
The left side of the entropy triangle corresponds to no deviation from the uniform distribution. The interactions of a network located at this side are uniformly distributed. Therefore, the potential freedom of choice is maximal. The bottom side of the entropy triangle corresponds to no mutual information between the interaction levels. The bottom species convey no information regarding the top species and *vice versa*. This indicates that there is no specialisation in the network. Finally, the right side of the triangle corresponds to no variation of information. There is no residual freedom of choice for the bottom and top species. Therefore, the stability of the network is low. The location of a network on the triangle gives us information regarding the importance of the different components of the balance equation and, hence, the interaction network structure. Networks that are located close to each other on the triangle will have a similar structure.

Three fictive interaction networks with extreme distributions are added to the triangle shown in Figure 4 in order to illustrate the use of the balance equation and the entropy triangle. These three extreme situations correspond to the three vertices of the triangle. To ease the interpretation, they are presented as interactions between anemone species and fish species. Table 2 contains the corresponding incidence matrices and their information-theoretic decomposition. Note that the presented matrices are binary matrices. The observations need to be converted to probabilities before information theory can be applied.

The upper vertex of the triangle shown in Figure 4 represents a network with a uniform distribution. Its variation of information is zero, while the mutual information is maximal. This situation corresponds to the left incidence matrix presented in Table 2, which is an example of perfect specialisation. Each fish species interacts with one specific anemone species and *vice versa*. The mutual information between the anemone species and fish species is maximal. If we know which fish species participated in an interaction, then we immediately know which anemone species was visited, as there is only one possibility. Similarly, if we know which anemone species was visited, then we immediately know the interacting fish species. Knowing the fish species reduces the uncertainty regarding the anemone species completely and knowing the anemone species reduces the uncertainty about the fish species completely. Therefore, the variation of information is equal to zero. There is no residual uncertainty and, thus, no freedom of choice. Such a network is maximally efficient, but vulnerable, since the limitations on possible interactions between the bottom and top species are very strict. In the absence of its specific anemone species, a fish species has no symbiotic partner. Because both marginal distributions are uniform, the deviation from the uniform distribution is zero.

The bottom-right vertex represents a network deviating maximally from the uniform distribution, while the mutual information and variation of information are zero. These characteristics correspond to the middle incidence matrix shown in Table 2, where one interaction is dominating the network. The variation of information is again equal to zero, but the mutual information is now also zero. Knowing the anemone species does not further reduce the uncertainty regarding the fish species, since there is simply no uncertainty, as there is only one possible interaction. However, the deviation from the uniform distribution is maximal, since both marginal distributions deviate completely from the uniform distribution as one interaction dominates the network.

Finally, the bottom-left vertex represents a network with no mutual information between the interaction levels and a maximal variation of information, while the deviation from the uniform distributions is zero. Therefore, freedom of choice is maximal. The right incidence matrix that is presented in Table 2 is an example of such a network, where each fish species interacts with every anemone species. The network is homogeneous, without any specialisation. Similar to the first incidence matrix, the deviation from the uniform distribution is equal to zero. However, the mutual information is now also zero. Knowing the anemone species does not reduce the uncertainty regarding the fish species, since every interaction is equally possible. On the other hand, the variation of information is maximal. In contrast to the left incidence matrix, this network has high stability. In the absence of one or even multiple anemone species, a fish species has plenty of other interaction options. The freedom of choice of the anemone species and fish species is not restricted at all. However, the network has a low efficiency as a result of the trade-off between stability and efficiency.

In Figure 4, the real interaction network between the anemone species and fish species is also added to the entropy triangle. The network is located very close to the left side of the triangle, which indicated that the deviation from the uniform distribution is minimal. Its structure lies somewhere in between the homogeneous network structure where each fish species interacts with every anemone species and the perfectly specialised network where each fish species interacts with one specific anemone species, but is slightly closer to perfect specialisation. A visual comparison of the interaction networks that are shown in Figure 4 supports this result. Figure 3 displays the entropy triangles of the joint distribution and the marginal distributions of this example. The black dot represents the real interaction network between anemone species and fish species. The three extreme interaction networks still correspond to the same three vertices of the triangle. Their location on the marginal triangles is the same as on the joint triangle because the marginal distributions of the bottom and top interaction level are identical in each network. Note that this is not always true and it entirely depends on the structure of the network. The location of the real interaction network is slightly different in the three triangles, but is still very similar. The marginal distribution of the top species deviates slightly more from the uniform distribution than the marginal distribution of the bottom species.

As demonstrated above, the balance equation indicates a trade-off between efficiency and stability: one comes at the cost of the other [39,43]. For example, Gorelick et al. [44] used entropy and mutual information to quantify the division of labour. Their method is similar to the information-theoretic decomposition described above and the subsequent normalisation in the entropy triangle. When species have a wider variety of interaction partners, their freedom of choice becomes larger. Therefore, the overall network stability increases [32], but the efficiency of the interactions decreases as they are less specialised [44]. Figure 5 illustrates this antagonistic relation. In this graph, the deviation from the uniform distribution is assumed to be constant. Therefore, the joint entropy of the network and, thus, the diversity of the network, remains constant. The variation of information increases with an increasing freedom of choice at the expense of the contribution of the mutual information. In a changing environment, stability will be an essential network characteristic. However, in a stable environment, efficiency will be a key factor [39]. The same graph can be constructed for the marginal distributions of the interaction levels, based on the decomposition of the balance equation into the separate contributions of the interaction levels. Table 3 summarises the elements of the information-theoretic decomposition of an interaction network and their ecological interpretation. Example networks are added in order to aid the interpretation.

Vázquez et al. [45] list several mechanisms that could explain the structure of an interaction network. The influence of these mechanisms can be linked to the components of the balance equation. The first mechanism, interaction neutrality, causes all individuals to have the same interaction probability. For binary incidence matrices, where the frequencies are not taken into account, this situation corresponds to a uniform distribution of the interactions and, thus, $H(U_{BT})$, the left-hand side of the balance equation. Other mechanisms will influence the distribution of the interactions and, therefore, influence the individual contributions of the three components at the right-hand side of the balance equation. Trait matching, for example, results in some interactions being favoured, while other interactions are impossible. The mutual information will increase as interactions become more efficient. However, as a result of the trade-off, the variation of information and stability will decrease as interactions are restricted. The spatio-temporal distribution will also influence the interactions. Species cannot interact if they are not at the same location at the same time. This can also be taken into account in the information-theoretic decomposition. Location, as well as time, can be introduced as an additional variable, in addition to the bottom and top interaction levels *B* and *T*. It will impose a further restriction on the interactions, leading to increased mutual information and a decrease in the variation of information. This notion will be discussed in Section 2.4. As mentioned before, Vázquez et al. [45] also note that observed interaction networks do not always match the true interactions due to sampling artefacts. Therefore, sampling can also influence the observed interaction structure and information-theoretic decomposition.

### 2.3. Entropy and Diversity

A proper interpretation of the information-theoretic indices is vital in correctly analysing ecological interaction networks. Entropy can be used to quantify the diversity of a network, but it is solely an index of diversity. Entropy is by no means synonymous for diversity [19,25]. For example, an ecosystem with a Shannon–Wiener index of four-bits is not twice as diverse as an ecosystem with a Shannon–Wiener index of two bits. The first system is four times as diverse, due to the logarithmic scale of the index. Similarly, a change in an interaction network will have a different effect on the diversity than on the entropy. More specifically, the order of magnitude of the effect will not be the same. The entropy of an incidence matrix with twice as many interactions will not be twice as high, because entropy does not obey the replication principle or doubling property [18,22]. This hampers the direct interpretation of information-theoretic indices. Converting entropies into *effective numbers* solves this problem. In this way, information-theoretic indices can be easily interpreted and compared, as they are now expressed on a linear scale. Indices are then no longer expressed in bits, but in the original units, i.e., the number of species or interactions, aiding the interpretation [8,21].

The *effective number of interactions* $E_{BT}$ is the number of interactions between *B* and *T* in a network with the same joint entropy, but with all species engaging in equally strong interactions. When all of the $E_{BT}$ interactions in a network are equally strong, the probability of an interaction $P_{ij}$ is equal to $\frac{1}{E_{BT}}$. Therefore, the effective number of interactions can be computed as
(31)$$H\left(B,T\right)=-\sum_{i=1}^{E_{BT}}\frac{1}{E_{BT}}\log\frac{1}{E_{BT}}=\log E_{BT}\;,$$
i.e.,
(32)$$E_{BT}=2^{H(B,T)}\;.$$
Note that, in the case of a binary incidence matrix, this results in the original number of interactions, as, without any information regarding the frequency or strength of the interactions, interactions were already assumed to be equally strong. For weighted matrices, the effective number of interactions will be different from the original number of interactions and will often not be an integer. Although the effective number of interactions might seem to be redundant for binary interaction matrices, it is not only applicable to the joint entropy, but also to other information-theoretic indices, where it is also more effective for binary interactions.

Similarly, we can define the marginal effective numbers $E_{B}=2^{H(B)}$ and $E_{T}=2^{H(T)}$, which represent the effective numbers of the bottom and top species, respectively. The conditional entropies give rise to $E_{B|T}=2^{H(B|T)}$ and $E_{T|B}=2^{H(T|B)}$, which, in turn, represent the average effective number of interactions for the top, resp. bottom, species.

Finally, the mutual information gives rise to the effective number $E_{I}=2^{I(B;T)}$, which represents the effective number of specific interactions. Together, the effective numbers give rise to:(33)$$\begin{array}{cc} E_{B} & =E_{I}E_{B|T} \end{array}$$(34)$$\begin{array}{cc} E_{T} & =E_{I}E_{T|B} \end{array}$$(35)$$\begin{array}{cc} E_{BT} & =E_{T|B}E_{B|T}E_{I}=E_{B}E_{T|B}=E_{T}E_{B|T}\;. \end{array}$$
Here, the last one is the most revealing. It states that the effective number of interactions equals the product of: (i) the effective number of interactions of the bottom species (i.e., $E_{T|B})$, (ii) the effective number of interactions of the top species (i.e., $E_{B|T}$), and (iii) the effective number of specific interactions (i.e., $E_{I}$).

### 2.4. Higher-Order Diversity

So far, only information-theoretic indices for two variables, which represent the bottom and top interaction levels, have been considered. However, the formulas introduced above can be easily extended to three or more variables. In the case of three variables, the third discrete variable *Z* could represent an additional species level, but also a different influencing factor, such as the location of the interaction, the time, or an environmental variable. The *joint entropy* of three variables can be computed as
(36)$$H\left(B,T,Z\right)=-\sum_{i=1}^{n}\sum_{j=1}^{m}\sum_{k=1}^{q}P_{ijk}\log P_{ijk}\;.$$
Other information-theoretic measures can be extended by conditioning them on the third variable *Z*. For example, for the mutual information, we have
(37)I(B;T|Z)=H(B|Z)−H(B|T,Z).
In a similar way, we can compute D(B,T|Z) and V(B,T|Z) (for more information, see MacKay [37] and Cover and Thomas [38]). Note that, in our framework, we currently do not consider expressions, such as I(B;T;Z) and I(B|T;Z). Only conditioning on a single variable is allowed. Some information theorists provide an interpretation for such multivariate measures [46], although as of yet there does not seem to be a consensus. We leave the ecological interpretation of such measures for future work.

For instance, consider the case where *Z* represents the location. By including this third variable in the indices, the influence of location on the uncertainty can be accounted for. In this way, entropy can be used to quantify alpha, beta, and gamma diversity. Alpha diversity is defined at a local scale, at a particular site [22]. This can be expressed by the conditional entropy given that the location is known:(38)$$H_{\alpha }=H\left(B,T|Z\right)\;.$$
The alpha entropy $H_{\alpha }$ quantifies the remaining uncertainty regarding the interactions when their location is known. Beta diversity, on the other hand, expresses the differentiation between local networks [22]. Therefore, beta entropy is the reduction in uncertainty that results from learning the location [25]:(39)$$H_{\beta }=H\left(B,T,Z\right)-H\left(B,T|Z\right)\;.$$
Using the chain rule for entropy [38], it can be shown that $H_{\beta }$ is also equal to the marginal entropy H(Z) of the location. Gamma diversity is the total diversity of an entire region. Because there is no knowledge regarding the location and, hence, also no reduction in uncertainty, gamma entropy can be quantified as
(40)$$H_{\gamma }=H\left(B,T,Z\right)\;.$$
The relation between alpha, beta, and gamma entropy is given by:(41)$$H_{\alpha }+H_{\beta }=H_{\gamma }\;.$$
These entropies can also be converted to effective numbers in the same way as above to be able to easily compare the alpha, beta, and gamma entropies, and interpret them as measures of interaction diversity:(42)$$E_{\alpha }=2^{H_{\alpha }}\;,\;E_{\beta }=2^{H_{\beta }}\;\;\mathrm{and}\;E_{\gamma }=2^{H_{\gamma }}\;.$$
By converting these entropies to effective numbers, the relation between alpha, beta, and gamma diversity, as proposed by Whittaker [47], is retrieved:(43)$$H_{\alpha }+H_{\beta }=H_{\gamma }\;⟺\;2^{H_{\alpha }}\;2^{H_{\beta }}=2^{H_{\gamma }}\;⟺\;E_{\alpha }\;E_{\beta }=E_{\gamma }\;\Leftrightarrow \;E_{\beta }=\frac{E_{\gamma }}{E_{\alpha }}\;.$$
Beta diversity can be quantified as the ratio between regional (i.e., gamma) and local (i.e., alpha) diversity [48].

The effective numbers have an interesting interpretation. $E_{\gamma }$ corresponds to the effective number of interactions over the networks, while $E_{\alpha }$ represents the effective number of unique interactions *per* individual network. Subsequently, we can interpret $E_{\beta }$ as the effective number of unique networks.

Figure 6 presents two fictive incidence matrices for two different locations to illustrate the use of alpha, beta, and gamma entropy, and the conversion to effective numbers. The joint incidence matrix of the bottom interaction level, top interaction level, and location contains ten binary interactions. Therefore, the non-zero $P_{ijk}$ values are equal to $\frac{1}{10}$. Alpha, beta and gamma entropy can be computed using the formulas that are derived above. As mentioned earlier, the entropies do not obey the doubling property. Converting them to effective numbers eases the interpretation. Figure 6 presents the resulting values. $E_{\beta }$ indicates that the interactions in the entire region, comprising the two locations, are almost twice as diverse as the local interactions. Inferring this directly from the value of $H_{\beta }$ is less straightforward.

## 3. Illustrations on Species Interaction Networks

### 3.1. Interaction Datasets

The information-theoretic analysis developed in this paper is applied to the interaction networks available in the Web of Life database (http://www.web-of-life.es/ (accessed on 1 June 2021)). This database contains various interaction types, including 51 host-parasite (HP), four plant-herbivore (PH), 17 anemone-fish (AF), four plant-ant (PA), 148 pollination (PL), and 34 seed dispersal (SD) interaction networks across the world. Weighted incidence matrices have been converted to binary observation by mapping non-zero values to one. The 27 food webs that are available in the Web of Life database were not included, as we are focusing on bipartite networks.

We performed the analysis using the *EcologicalNetworks.jl* package in the Julia programming language [49], which we extended to include the indices that are described in this work. This package allows for an easy analysis of ecological interaction networks and it contains the information-theoretic indices introduced in this work, in addition to various other non-information-theoretic features for ecological network analysis. All of the Web of Life networks can also be accessed via this package. From here on, when we talk about the normalized components, either global or for a trophic level, we will informally use the terms D-, I-, and V-component.

### 3.2. Web of Life Interaction Networks

Figure 7 shows the information-theoretic decomposition of the interaction networks of the Web of Life database in the triangle entropy plot, while Figure 8 shows an extract for the 16 anemone–fish interaction networks included. There is no clear difference between the decompositions of the different interaction types. More striking is the fact that all of the networks are located at the left side of the triangle. The full range of the V-component and I-component is used, while the range of the D-component is limited. This can be explained by the fact that all of the networks are binary. In binary networks, the deviation from the uniform distribution is limited. The larger the network, the larger the deviation can be, but the larger $H(U_{BT})$ will be, the denominator of the normalised balance equation. We recall that any weighted incidence matrix was binarised.

Figure 9 was created using the original data, containing both binary and weighted incidence matrices, in order to illustrate the full potential of the entropy triangle. This figure shows that, when weighted observations are used, the entire range of the D-component and, therefore, the entire triangle, is utilised. However, this does not mean that the information-theoretic decomposition of binary incidence matrices is irrelevant. Contrary to the D-component, the V-component and I-component show a wide spread for binary incidence matrices. These two components precisely explain the ecologically important trade-off between stability and efficiency. Because the D-component is small and less essential to analyse the network structure of binary networks, it makes sense to convert the entropy triangle to a graph with only two axes, one for the V-component and one for the I-component. Figure 10 shows the relation between the normalised V- and I-components. The values are the same as in Figure 7, with the only difference being that the D-component is no longer shown. Because the D-component is small, Figure 10 shows a linear relation, once more illustrating the trade-off between stability and efficiency.

### 3.3. Higher-Order Diversity: Time and Space

This section illustrates how higher-order information-theoretic indices can be used to study a spatially distributed metaweb. To this end, we use the networks that were collected by Hadfield et al. [50], containing 51 host-parasite networks collected over a large spatial region in Eurasia. The aggregated metaweb contains 206 flea species and 121 mammal species.

The collection of networks could be represented as a 206×121×51 tensor, where the respective dimensions correspond to flea species, mammal species, and location. We obtained the trivariate probability mass function by normalising this presence–absence tensor. Subsequently, we computed the higher-order indices, as described in Section 2.4, as well as the information-theoretic decomposition, both marginalised as well as conditional, e.g., I(B;T) and I(B;T|Z). Table 4 presents these results.

The alpha, beta, and gamma entropy are presented both in bits and in equivalent effective numbers. It is notable that the former are additive, while the latter are multiplicative. Here, because all of the interactions are equally weighted, $E_{\gamma }$ boils down to the total number of interactions distributed over all networks. This number is split into two parts, where $E_{\alpha }$ can be seen as an estimate of the ’average’ number of unique interactions for each network. At the same time, $E_{\beta }$ corresponds to the theoretical number of unique networks. Because of the overlap of the interactions, $E_{\beta }$ is substantially smaller than the 51 networks that were observed.

In the second part of Table 4, we present the information-theoretic decomposition. The marginal version corresponds to the setting where we have summed over the location variable *Z*. In contrast to many other networks that are discussed in this work, this marginal network does not have equal probabilities for all present interactions, since the same ecological interaction can occur multiple times at different locations. The V-component seems to dominate this metaweb, meaning that this metaweb has plenty of redundancy for the species, which implies stability. The D-component is relatively low, which means that the species composing the network have a relatively equal importance. However, the marginal indices take the differences between networks into account, which leads to a completely different picture. The D-component is now dominating, which suggests that individual networks are largely determined by marginals or activities of the composing species. Conditioning reduces the relative importance of the I-component, showing that specificity is more present in the metaweb when compared to the individual networks. Note that, in whatever way the decomposition is computed, its components sum to 14.60≈log(206×121), as suggested by the theory.

## 4. Discussion and Conclusions

Our work translates species interaction networks into bivariate distributions and it gives meaning to the various information-theoretic measures that one can compute. As discussed in the introduction, we are most certainly not the first (nor will we be the last) authors to apply information theory to community ecology. However, our work introduces the elegant balance equation of Valverde-Albacete and Peláez-Moreno [42], allowing for researchers to decompose the information in an interaction network. Similarly, we find that the information also decomposes interestingly across spatial and temporal scales, in a way that is compatible with the concepts of alpha, beta, and gamma diversity.

Normalising an incidence matrix results in a valid probability distribution. In this work, we mainly used presence–absence, basically giving every interaction an equal weight. This process removes considerable information regarding the interaction strength that is present in the visitation rates and similar data. Although the frequency of interaction does not always match the interaction strength [3,51], it can convey information regarding the relative importance of the species [52]. We found directly interpreting the abundances as frequencies to be suboptimal, both conceptually and empirically. Nevertheless, using normalised binary incidence matrices has the drawback of giving equal weight to all interactions, while it gives no weight to the possible interactions that were not sampled. These are chiefly limitations of the data collection side and not of the theoretical framework.

The problems outlined above arise because the sampled network is a realisation of the underlying interaction probabilities rather than the probabilities themselves. The challenge is to find a suitable statistical method for recovering these probabilities from the observations. Here, thresholding and normalising are straightforward ones.

Several authors have proposed more sophisticated models for recovering interaction probabilities, for example, based on Bayesian reasoning [29,53,54] or filtering [26,55]. In recent work, some of the present authors suggested a MaxEnt model to recover interaction probabilities given species abundances and interaction preferences [31]. Finding suitable methods for estimating the various proposed indices and assessing their statistical relevance is an important future challenge.

The proposed framework does not make use of the species identities. Indeed, all of the indices are invariant under permutation of rows and columns. As such, they do not take some species potentially having a similar ecological function into account. Leinster and Cobbold [56] addressed this problem by incorporating species similarities (for example, based on traits) as weights into the Shannon entropy. Recent work that was conducted by Gallego et al. [57] generalised this approach to all information-theoretic measures, including mutual information and a variation of information, to study probability distributions over any metric space. Thus, the latter work is fully compatible with our proposed methodology for analysing species interaction networks. Combining the proposed information-theoretic framework with the ecological role of the species will result in a much more in-depth characterization of an ecosystem.

When Shannon and Weaver popularised the landmark paper *The Mathematical Theory of Communication* to the non-specialist, they clarified that information is fundamentally related to the freedom of choice [14]. We interpreted this as an ecological choice, where we quantify the freedom of the top species given the bottom species and *vice versa*.

## Figures and Tables

**Figure 1 entropy-23-00703-f001:**
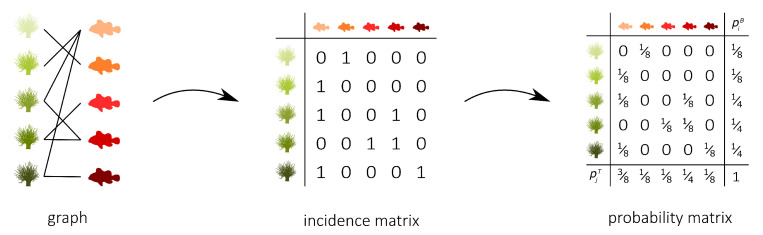
The interactions between five anemone species and five fish species [11] represented as a bipartite graph (**left**), binary incidence matrix (**middle**), and probability matrix of the joint distribution with an indication of the marginal probabilities of the anemone species $p_{i}^{B}$ and the fish species $p_{j}^{T}$ (**right**). This example will serve throughout this work to illustrate the proposed indices.

**Figure 2 entropy-23-00703-f002:**
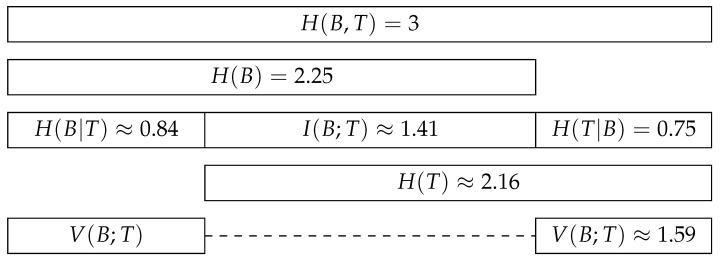
The visualisation of the information-theoretic decomposition of the interaction network between five anemone species and five fish species, as presented Figure 1. Image after MacKay [37].

**Figure 3 entropy-23-00703-f003:**
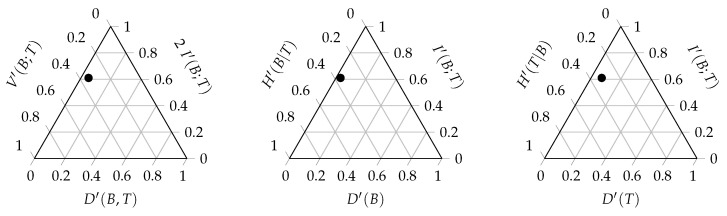
The triangle entropy plot of the total balance equation (**left**) and the marginal balance equations of the bottom (**middle**) and top (**right**) interaction level. The black dot represents the interaction network between five anemone species and five fish species, as presented in Figure 1.

**Figure 4 entropy-23-00703-f004:**
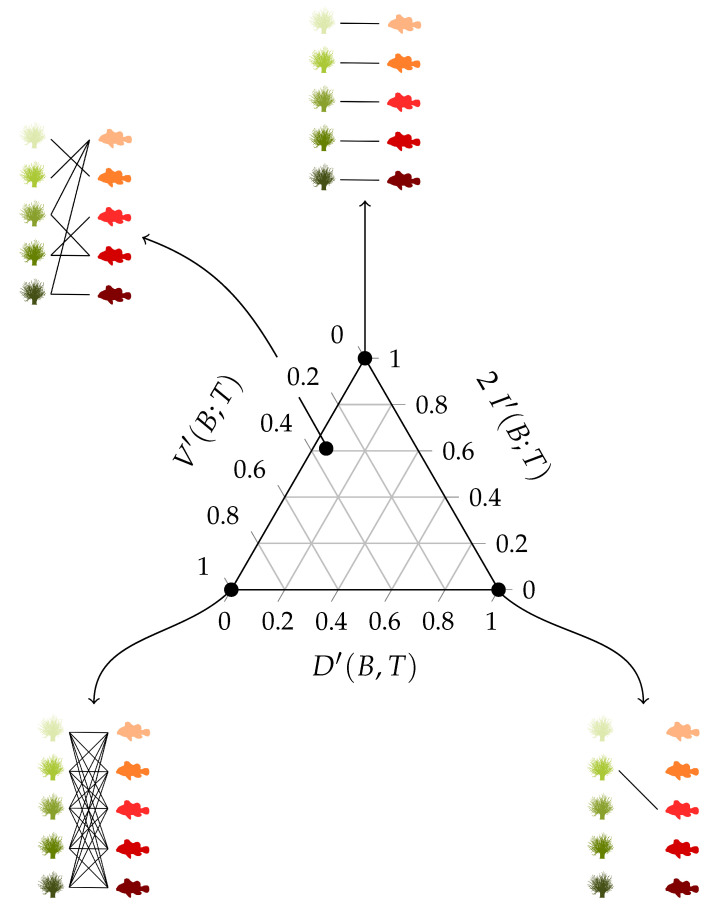
Triangle entropy plot with three fictive interaction networks being located at the vertices of the triangle and the real interaction network between five anemone species and five fish species to illustrate the interpretation of the balance equation. The corresponding incidence matrices are presented in Table 2.

**Figure 5 entropy-23-00703-f005:**
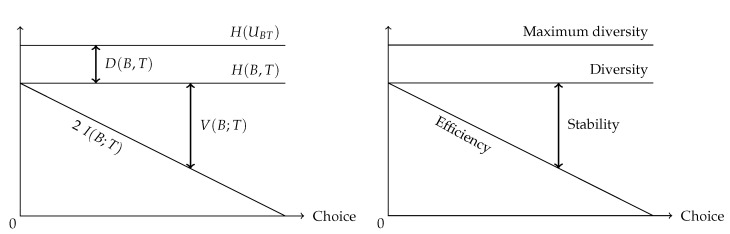
A graphical representation of the relation between joint entropy, mutual information, and variation of information (**left**) to illustrate the trade-off between stability and efficiency (**right**). Image after Rutledge et al. [32].

**Figure 6 entropy-23-00703-f006:**
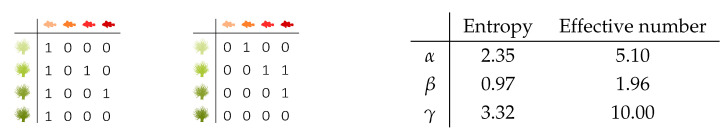
Fictive incidence matrices for two different locations to illustrate alpha, beta, and gamma entropy, and the conversion to effective numbers. There are 5.1 equivalent interactions per network and 1.96 equivalent networks.

**Figure 7 entropy-23-00703-f007:**
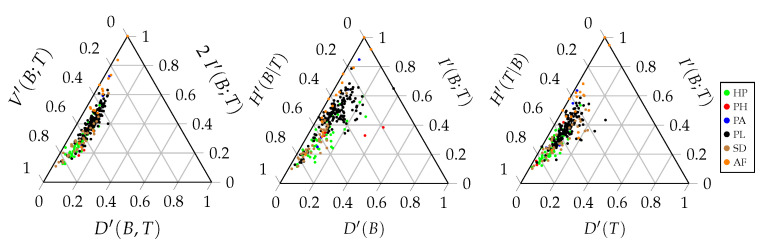
Triangle entropy plot showing the information-theoretic decomposition of the total balance equation **(left**) and the marginal balance equations of the bottom (**middle**) and top (**right**) interaction level. The different colours denote the interaction types as described in Section 3.1.

**Figure 8 entropy-23-00703-f008:**
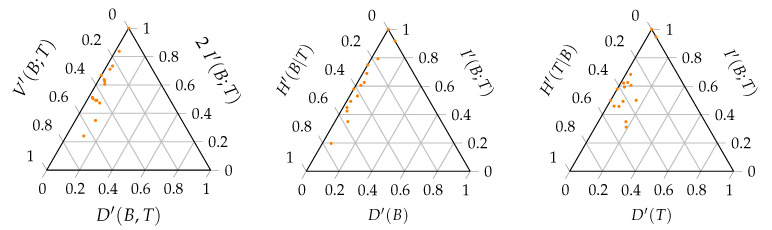
The triangle entropy plot showing the information-theoretic decomposition of the total balance equation (**left**) and the marginal balance equations of the bottom (**middle**) and top (**right**) interaction level of the 16 anemone–fish interaction networks that were observed by Ricciardi et al. [11].

**Figure 9 entropy-23-00703-f009:**
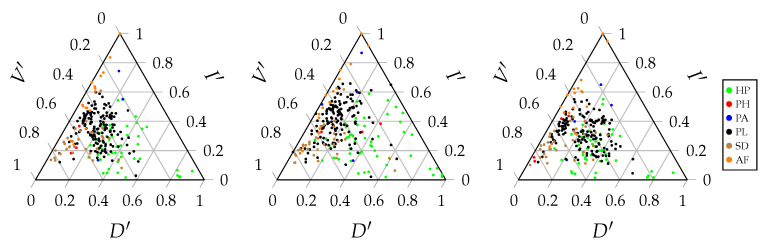
Triangle entropy plot showing the information-theoretic decomposition of the total balance equation (**left**) and the marginal balance equations of the bottom (**middle**) and top (**right**) interaction level of the *original* interaction networks, being described by both binary and weighted incidence matrices. The different colours denote the interaction types, as described in Section 3.1.

**Figure 10 entropy-23-00703-f010:**
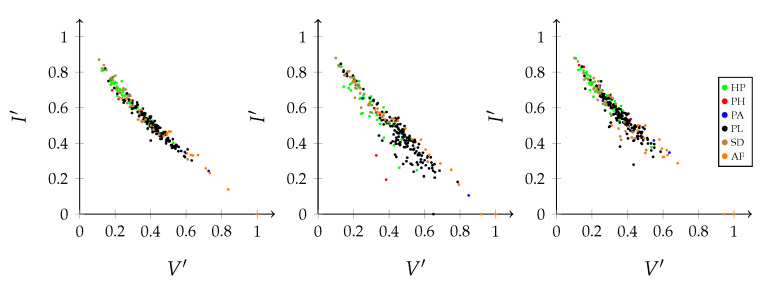
The relation between the normalised V- and I-components for the joint distribution (**left**) and the marginal distributions of the bottom (**middle**) and top (**right**) interaction level. The different colours denote the interaction types, as described in Section 3.1.

**Table 1 entropy-23-00703-t001:** An overview of the different components of the balance equation of the joint distribution and marginal distributions of the bottom and top species. The entropy of the uniform distribution is equal to the sum of the corresponding *D*, *I*, and *V* components.

	*H*	*D*	*I*	*V*
Joint	$H(U_{BT})$	D(B,T)	2I(B;T)	V(B;T)
Bottom level	$H(U_{B})$	D(B)	I(B;T)	H(B|T)
Top level	$H(U_{T})$	D(T)	I(B;T)	H(T|B)

**Table 2 entropy-23-00703-t002:** Fictive incidence matrices illustrating the interpretation of the balance equation. Figure 4 presents the triangle entropy plot with the corresponding interaction networks.

		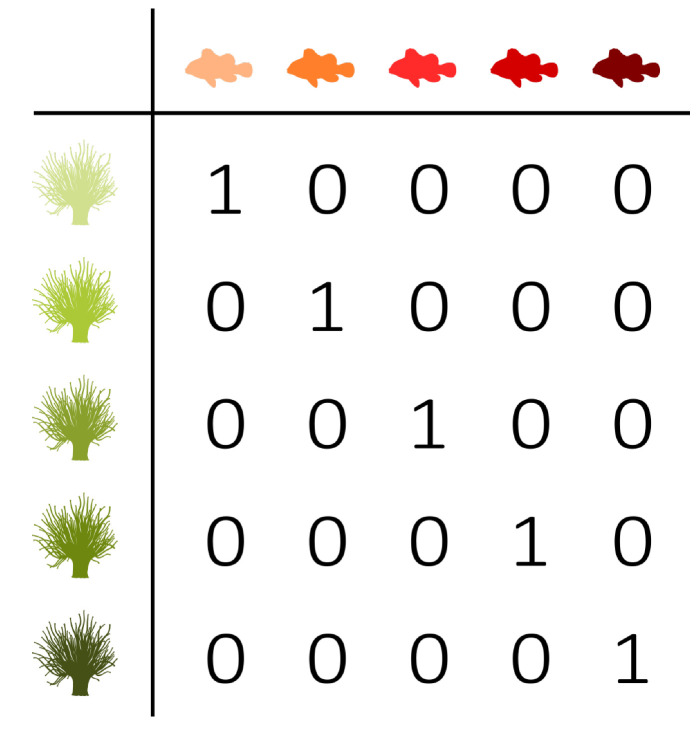	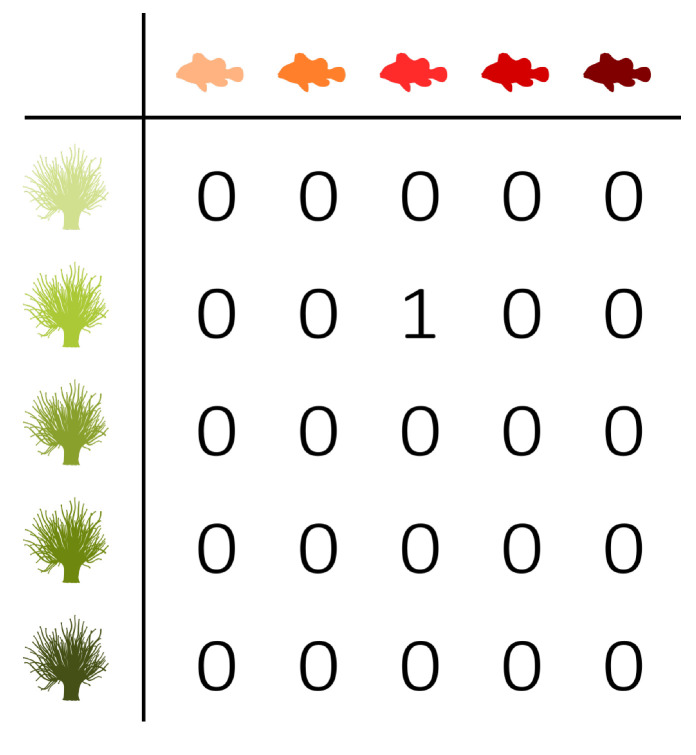	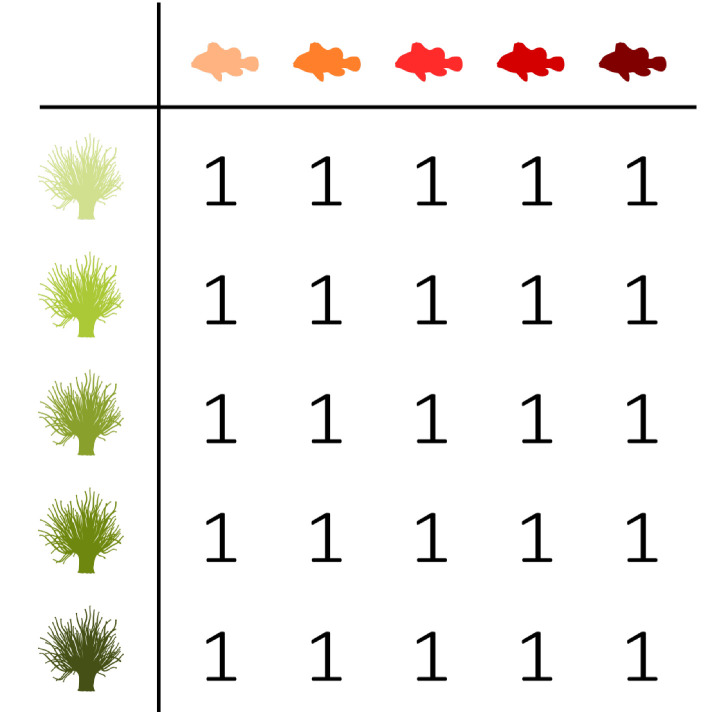
$D^{\prime }\left(B,T\right)$		0	1	0
$I^{\prime }\left(B;T\right)$		1	0	0
$V^{\prime }\left(B;T\right)$		0	0	1

**Table 3 entropy-23-00703-t003:** An overview of the components of the information-theoretic decomposition of an interaction network and their ecological interpretation. For $H(U_{BT})$, the last column contains a network with uniformly distributed interactions. For the other indices, the last column contains a network with a low (left) and high (right) value for the respective index.

Index	Ecological Interpretation	Example Networks
$H(U_{BT})$	Entropy of the network if all interactions would be uniformly distributed over the species and therefore the freedom of choice would be maximal.	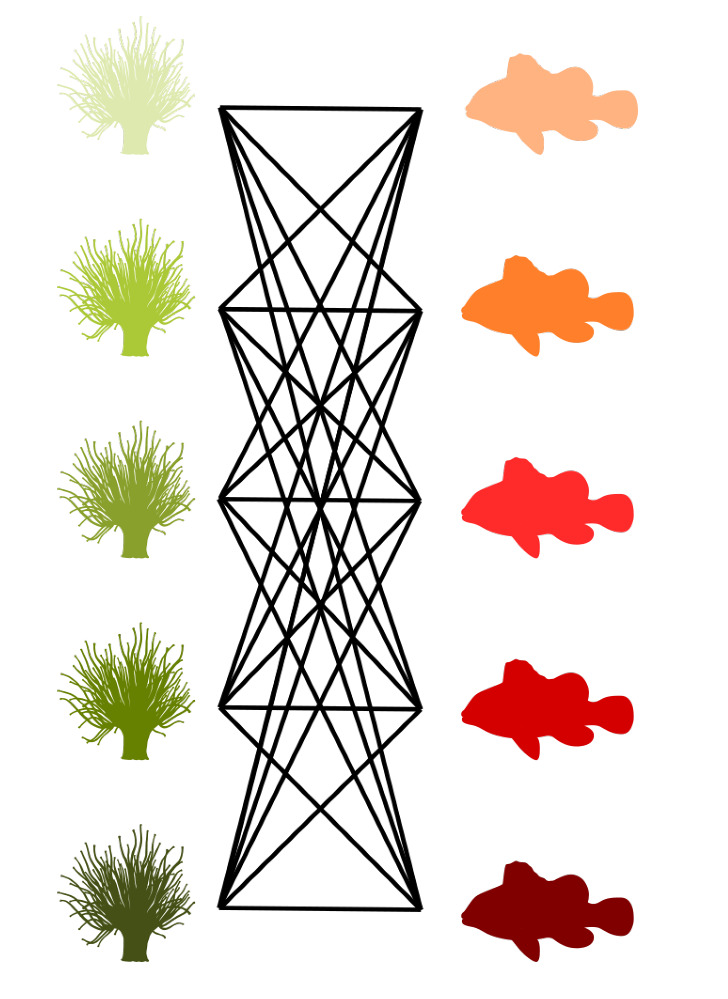
D(B,T)	Expresses how much the observed interactions differ from a uniform distribution. A large deviation indicates that one or more interactions dominate the network and that the freedom of choice is restricted.	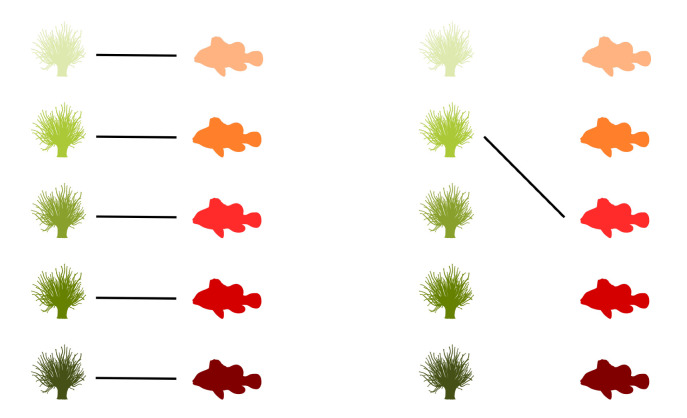
I(B;T)	Quantifies the level of organisation of the network, i.e., the limitation on possible interactions between the bottom and top species. A restricted number of possible interactions, i.e., a large mutual information, can lead to a higher efficiency.	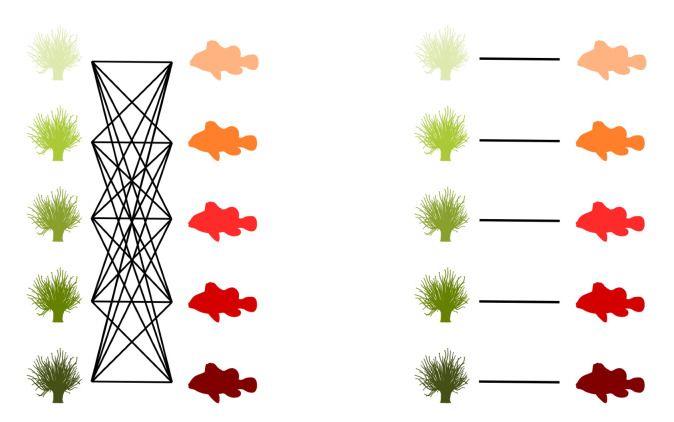
V(B;T)	Quantifies the uncertainty that remains when the structure of the interaction network is known. A large variation of information corresponds to a large variety of possible interaction partners and thus a large uncertainty. This index can be seen as a measure of the network’s stability. A restriction of the number of possible interactions and thus freedom of choice of the species, decreases the stability of the network.	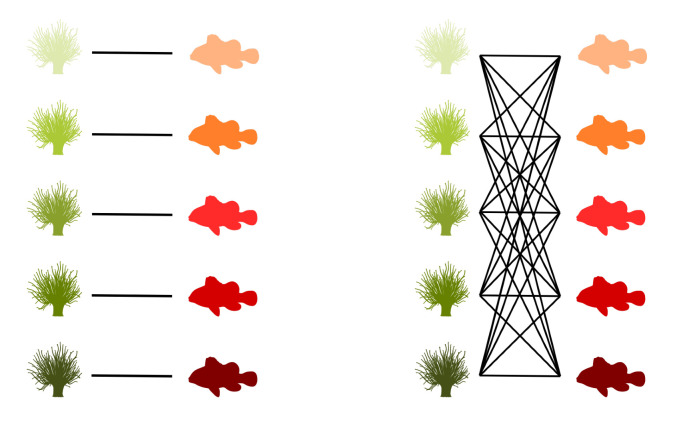
H(B|T) H(T|B)	Quantifies the uncertainty that remains about the bottom species when the top species are known or vice versa. A large conditional entropy indicates that the interacting species have a large freedom of choice. This index is similar to the variation of information described above, but is based on the marginal distribution of a single interaction level, whereas the variation of information combines the information of both marginal distributions. A low conditional entropy indicates that the freedom of choice of the species of the interaction level is restricted, lowering the stability of the network.	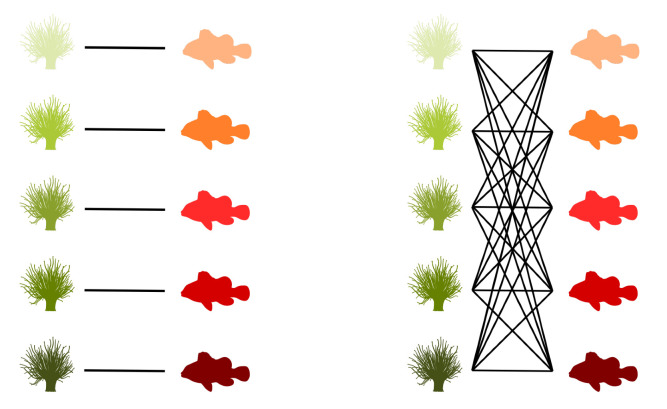

**Table 4 entropy-23-00703-t004:** Information–theoretic analysis of the host–parasite networks of Eurasia. (left) The alpha, beta, and gamma entropy and their corresponding effective numbers. (right) The marginal and conditional versions of the D-, I-, and V-components of the entropy decomposition.

	α	β	γ			*D*	2I	*V*
entropy *H*	6.569	5.391	11.960		marginal	1.917	4.128	8.561
effective number *E*	94.938	41.964	3984		conditional	6.940	2.194	5.472

## Data Availability

The data used for this study was retrieved from the Web of Life website and can be accessed via the EcologicalNetworks.jl package.

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
