# Peer review of "Disentangling the Information in Species Interaction Networks"

_entropy, 2021, doi:10.3390/e23060703_

Round 1
Reviewer 1 Report
Overview:
This paper presents a review of how information-theoretic measures can be used to quantify the abundance, specificity and the redundancy in ecological interaction networks, with a particular focus on a particular balance equation. The analysis appears to be sound and the work is very well-presented. With the exception of one issue, I believe that this article is suitable for publication in Entropy.
Reviewer context:
My background is in information theory and complex systems. I do not have a background in ecology. As such, my review mostly focuses on how information theory has been used to analyse the networks, rather than the significance of this work to ecologists. This is also why I did not comment of the Significance of Content or the Interest to the Readers above.
Major issue:
In Section 2.4, the authors state that "all other information-theoretic measures can be extended" to three or more variables. This is true. However, the multivariate mutual information (MMI) between three variables is not non-negative. (See MacKay, Chapter 8, "Dependent Random Variables", Section 8.4.) Given how the authors use this mutual information as an index that quantifies the level of organisation of the network, a negative value for I(B;T;X) seems to be quite problematic. The authors should either (1) provide some clarification regarding this potentially negative index, or (2) avoid claiming that we can extend the mutual information to more than two variables.
Just to provide the authors with some further context, the potential negativity of the MMI is a contentious issue in information theory. Csiszar and Korner, for example, argued that the MMI had "no intuitive meaning", while Cover and Thomas noted that, "there isn't really a notion of mutual information common to three random variables" (pg. 49 of my copy). Recently, Williams and Beer proposed a framework called partial information decomposition (PID) that seems to provide an explanation for this result. In short, a positive value of the MMI is associated with two variable providing the same redundant information about the third variable, while a negative value of the MMI is associate with two variables providing synergistic information about the target. However, PID is not universally accepted and there is currently a significant amount of ongoing research on the topic.
All of this is to say that the authors would probably do best by avoiding option (1). The subsequent parts of Section 2.4 which covers the alpha, beta and gamma diversity does not seem to rely on the MMI, so it seem to me that the authors could choose option (2) without harming their results. I have called this a major issue because further clarification is required, but I think it only requires a relatively minor change to achieve this clarity.
Csiszar and Körner. Information Theory: Coding Theorems for Discrete Memoryless Systems. Academic Press Inc.: Cambridge, MA, USA, 1981.
PL Williams, RD Beer. Nonnegative decomposition of multivariate information. ArXiv preprint arXiv:1004.2515
Minor issues:
(1) Lines 2 and 18: The authors often refer to the information-theoretic measures as indices. From my perspective, this is a little unusual. (Perhaps this is how they are typically used in ecology?) I think a comment or slight rephrasing in the introduction would help. It is already kind of there in the paragraph beginning line 18, but I think it needs to be flipped; i.e. the information-theoretic measures can be used as indices that quantify different network structure.
(2) Line 23: "Graphs are a common representation of interaction networks". This reads a little like "Networks are commonly used to represent networks". I think you mean something more like "Networks are commonly used to represent ecological interactions".
(3) Line 49: You risk upsetting/confusing physicists by mentioning (Shannon) entropy beside the word energy, which is more typically associated with thermodynamic entropy. Consider rewording.
(4) Line 54: the "information content" h(x) = - log p(x) is a named function (see MacKay) that quantifies the information content of a single event x. The entropy is then the expected information content over all events x from the variable X, i.e. H(X) = E_X[h(x)] = - \sum_x p(x) log p(x). As you have written it, these functions seem to be confused. Perhaps reword here to avoid confusion.
(5) Lines 207-211: I think D(B,T) could be a little better explained. The product distribution P(X*Y) of two marginals P(X) and P(Y) is the joint distribution that is generated by assuming that the marginals are independent, i.e. P(X*Y) = P(X)*P(Y). When we consider the entropy of this product distribution, we get H(X*Y) = H(X) + H(Y), which is a consequence of the property of the logarithm. Thus, the function D(B,T) is the difference between the maximal entropy joint
distribution H(U_BT) and the entropy of the product distribution H(B*T) H(B) + H(T).
(6) Line after 218 (line number is missing): You should probably provide a citation for de Finetti diagrams for us non-ecologists, as I assume is a standard thing in ecology. In any case, I am more concerned with de Finetti entropy diagrams: is this something new that you are introducing here? If so, please state that explicitly as it is not clear. If not, then please provide a citation.
(7) Line 285: You seem to be saying that the result in Fig 4 corresponds to the example given in Fig 3, and perhaps even Fig 1 also? I think this could be clarified. You should also consider stating in the caption of Fig 1 that, "this example will be used again later in the paper" (I know you can't refer forwards to Figs 3 and 4 in the caption of Fig 1).
(8) Table 3: Regarding the example networks. They mostly occur in pairs, it seems, with one corresponding to a low value and the other to a high value. If so, this should clarified somewhere.
(9) Section 2.3, second to ninth lines (line numbers are missing): This section is a little confusing to me. It seems to me that what you are saying is that the information-theoretic measure that you have defined may not always align with your intuitive notion of the level of diversity. The information-theoretic measure of diversity tells you how many bits of information it takes to specify an interaction on average. If you were to split each interaction into two equally likely sub-interactions (by, say, splitting each fish species into two equally likely sub-species), then the information-theoretic measure of diversity would increase by 1 bit. This is, of course, what this measure was designed to do (see Shannon's derivation of the entropy in "A Mathematical Theory of Communication"), and as such it is a little odd to expect it to do otherwise. I think your point is that you do need to be careful because its not a simple index that will scale linearly with your intuitive preconceptions of diversity. However, it kind of reads like you are criticising the measure for failing to do something that it was never designed to do, which is a little silly. Perhaps you should consider rewording it slightly. (Of course, you should also feel free to ignore me on this point, as I appreciate that this comment was not targeted at information-theorists like me.)
(10) Line 448: This first line of this section seemed very unusual and off-topic compared to the second.
Author Response
Overview:
This paper presents a review of how information-theoretic measures can be used to quantify the abundance, specificity and the redundancy in ecological interaction networks, with a particular focus on a particular balance equation. The analysis appears to be sound and the work is very well-presented. With the exception of one issue, I believe that this article is suitable for publication in Entropy.
Reviewer context:
My background is in information theory and complex systems. I do not have a background in ecology. As such, my review mostly focuses on how information theory has been used to analyse the networks, rather than the significance of this work to ecologists. This is also why I did not comment of the Significance of Content or the Interest to the Readers above.
Major issue:
In Section 2.4, the authors state that "all other information-theoretic measures can be extended" to three or more variables. This is true. However, the multivariate mutual information (MMI) between three variables is not non-negative. (See MacKay, Chapter 8, "Dependent Random Variables", Section 8.4.) Given how the authors use this mutual information as an index that quantifies the level of organisation of the network, a negative value for I(B;T;X) seems to be quite problematic. The authors should either (1) provide some clarification regarding this potentially negative index, or (2) avoid claiming that we can extend the mutual information to more than two variables.
Just to provide the authors with some further context, the potential negativity of the MMI is a contentious issue in information theory. Csiszar and Korner, for example, argued that the MMI had "no intuitive meaning", while Cover and Thomas noted that, "there isn't really a notion of mutual information common to three random variables" (pg. 49 of my copy). Recently, Williams and Beer proposed a framework called partial information decomposition (PID) that seems to provide an explanation for this result. In short, a positive value of the MMI is associated with two variable providing the same redundant information about the third variable, while a negative value of the MMI is associate with two variables providing synergistic information about the target. However, PID is not universally accepted and there is currently a significant amount of ongoing research on the topic.
All of this is to say that the authors would probably do best by avoiding option (1). The subsequent parts of Section 2.4 which covers the alpha, beta and gamma diversity does not seem to rely on the MMI, so it seem to me that the authors could choose option (2) without harming their results. I have called this a major issue because further clarification is required, but I think it only requires a relatively minor change to achieve this clarity.
Answer: We thank the reviewer for pointing out the recent work on multivariate mutual information. In our work, we follow the framework outlined by MacKay where expressions as I(B;T;Z) and I(Z|B,T) are illegal. We only consider I(B;T|Z). Hence, we fully agree with the statement of the reviewer. We should have made this clearer and have remedied this in the text. Section 2.4 only touches upon higher-order IT measures briefly, mainly because the link with alpha-, beta- and gamma-diversity is so nice. It likely deserves a more detailed investigation (discussing the sense of MMI) in future work.
Csiszar and Körner. Information Theory: Coding Theorems for Discrete Memoryless Systems. Academic Press Inc.: Cambridge, MA, USA, 1981.
PL Williams, RD Beer. Nonnegative decomposition of multivariate information. ArXiv preprint arXiv:1004.2515
Minor issues:
(1) Lines 2 and 18: The authors often refer to the information-theoretic measures as indices. From my perspective, this is a little unusual. (Perhaps this is how they are typically used in ecology?) I think a comment or slight rephrasing in the introduction would help. It is already kind of there in the paragraph beginning line 18, but I think it needs to be flipped; i.e. the information-theoretic measures can be used as indices that quantify different network structure.
Answers: We had several internal discussions about this. Mathematically, “measure” is the correct term. However, ecologists mainly use “indices”, for example, to quantify diversity (cfr., index is in fashion in economics as well). So, we often use “index” as an ecologist would use this term. We added a footnote in the introduction to clarify this and went over the examples. As we aim to have interdisciplinary work, we still use both terminologies (at the risk of not being 100% satisfactory to both communities).
(2) Line 23: "Graphs are a common representation of interaction networks". This reads a little like "Networks are commonly used to represent networks". I think you mean something more like "Networks are commonly used to represent ecological interactions".
Answer: This was a clumsy sentence we modified.
(3) Line 49: You risk upsetting/confusing physicists by mentioning (Shannon) entropy beside the word energy, which is more typically associated with thermodynamic entropy. Consider rewording.
Answer: The study investigated the distribution of energy and used Shannon’s theory. So the sentence describes the study as it is.
(4) Line 54: the "information content" h(x) = - log p(x) is a named function (see MacKay) that quantifies the information content of a single event x. The entropy is then the expected information content over all events x from the variable X, i.e. H(X) = E_X[h(x)] = - \sum_x p(x) log p(x). As you have written it, these functions seem to be confused. Perhaps reword here to avoid confusion.
Answer: This was a bit sloppy on our side, expected information content makes it correct.
(5) Lines 207-211: I think D(B,T) could be a little better explained. The product distribution P(X*Y) of two marginals P(X) and P(Y) is the joint distribution that is generated by assuming that the marginals are independent, i.e. P(X*Y) = P(X)*P(Y). When we consider the entropy of this product distribution, we get H(X*Y) = H(X) + H(Y), which is a consequence of the property of the logarithm. Thus, the function D(B,T) is the difference between the maximal entropy joint
distribution H(U_BT) and the entropy of the product distribution H(B*T) H(B) + H(T).
Answer: This is a good suggestion, as D(B,T) was a bit harder to explain clearly. We have emphasized this part in our revision.
(6) Line after 218 (line number is missing): You should probably provide a citation for de Finetti diagrams for us non-ecologists, as I assume is a standard thing in ecology. In any case, I am more concerned with de Finetti entropy diagrams: is this something new that you are introducing here? If so, please state that explicitly as it is not clear. If not, then please provide a citation.
Answer: It is just a ternary diagram used to represent vectors in the 2-simplex. Since the De Finetti diagram is likely unnecessarily confusing (and is actually more used in population genetics), we have replaced it with ‘triangle entropy plot.
(7) Line 285: You seem to be saying that the result in Fig 4 corresponds to the example given in Fig 3, and perhaps even Fig 1 also? I think this could be clarified. You should also consider stating in the caption of Fig 1 that, "this example will be used again later in the paper" (I know you can't refer forwards to Figs 3 and 4 in the caption of Fig 1).
Answer: We have added this.
(8) Table 3: Regarding the example networks. They mostly occur in pairs, it seems, with one corresponding to a low value and the other to a high value. If so, this should clarified somewhere.
Answer: This is explained in the caption of the table.
(9) Section 2.3, second to ninth lines (line numbers are missing): This section is a little confusing to me. It seems to me that what you are saying is that the information-theoretic measure that you have defined may not always align with your intuitive notion of the level of diversity. The information-theoretic measure of diversity tells you how many bits of information it takes to specify an interaction on average. If you were to split each interaction into two equally likely sub-interactions (by, say, splitting each fish species into two equally likely sub-species), then the information-theoretic measure of diversity would increase by 1 bit. This is, of course, what this measure was designed to do (see Shannon's derivation of the entropy in "A Mathematical Theory of Communication"), and as such it is a little odd to expect it to do otherwise. I think your point is that you do need to be careful because its not a simple index that will scale linearly with your intuitive preconceptions of diversity. However, it kind of reads like you are criticising the measure for failing to do something that it was never designed to do, which is a little silly. Perhaps you should consider rewording it slightly. (Of course, you should also feel free to ignore me on this point, as I appreciate that this comment was not targeted at information-theorists like me.)
Answer: This section discusses a problem some ecologists have with using Shannon entropy to quantify biodiversity. Some authors erroneously interpret a system with H(X) = 4 to be twice as diverse as a system with H(X)=2. The error here is of course that entropy and the other measures live on a logarithmic scale. So we (and other work we refer to) suggest transforming the measures into effective numbers, making the units species or interactions. For example, the first system would be as diverse as if it contained 16 species, while the second system has an effective diversity of 4 species. This is a natural way to interpret the measures for diversity. For information-theorists, this is likely completely obvious though it is important to point out to other readership. The effective numbers lead to some interesting, non-trivial insights in this section and the next, so it is worth discussing them. We do agree that this section could be improved so we revised it to make it more clear.
(10) Line 448: This first line of this section seemed very unusual and off-topic compared to the second.
Answer: We might have gone a bit too meta here, so we removed this line.
Reviewer 2 Report
This paper’s contribution is to introduce some measures based on entropy to measure the species interaction networks. The paper can be accepted after MIJOR REVISION. Some minor points are given as follows. 1. All equations should be marked in the order of numbers. 2. In Page 4, $p_{j}^{T} = \sum\limits_{i=1}nP_{ij}$ should be corrected as $p_{j}^{T} = \sum\limits_{i=1}^{n}P_{ij}$ in Latex language, otherwise it not is right logically. 3. The introduction of measures like entropy, marginal entropy and so forth, is too clumsy and it should be shortened. 4. Another round of proofreading is needed. For example, in Page 18, “giving equal weight to all interactions” should be modified “giving equal weights to all interactions”. 5. What is the future research direction of this work? Authors should give a discussion at the end of this work.Author Response
This paper’s contribution is to introduce some measures based on entropy to measure the species interaction networks. The paper can be accepted after MIJOR REVISION. Some minor points are given as follows.
- All equations should be marked in the order of numbers.
Answer: The reviewer is correct, this is now updated according to the style of Entropy.
- In Page 4, $p_{j}^{T} = \sum\limits_{i=1}nP_{ij}$ should be corrected as $p_{j}^{T} = \sum\limits_{i=1}^{n}P_{ij}$ in Latex language, otherwise it not is right logically.
Answer: Fixed, thanks for pointing out.
- The introduction of measures like entropy, marginal entropy and so forth, is too clumsy and it should be shortened.
Answer: We agree that these sections are quite long, though we want to both build this framework from the ground up for readers not familiar with information theory as well as link it to the ecological concepts for those who do. Our desire to be didactic makes these sections quite extended.
- Another round of proofreading is needed. For example, in Page 18, “giving equal weight to all interactions” should be modified “giving equal weights to all interactions”.
Answer: We read the paper again, but we think both formulations are correct, giving weight (=importance) that is equal vs setting all the weights (parameters) to the same value.
- What is the future research direction of this work? Authors should give a discussion at the end of this work.
Answer: in the discussion section, we outline two points of future research (with pointers to relevant literature): (1) estimation of the indices and (2) using traits and phylogeny of the species. We have reexamined these parts to make them more clear.